# A Cross-Sectional Study of the Dermatological Manifestations of Patients with Fabry Disease and the Assessment of Angiokeratomas with Multimodal Imaging

**DOI:** 10.3390/diagnostics13142368

**Published:** 2023-07-14

**Authors:** Pálma Anker, Luca Fésűs, Norbert Kiss, Anna Lengyel, Éva Pinti, Ilze Lihacova, Alexey Lihachev, Emilija Vija Plorina, György Fekete, Márta Medvecz

**Affiliations:** 1Department of Dermatology, Venereology and Dermatooncology, Semmelweis University, 1085 Budapest, Hungary; anker.palma@med.semmelweis-univ.hu (P.A.); fesus.luca@med.semmelweis-univ.com (L.F.); kiss.norbert@med.semmelweis-univ.hu (N.K.); 2Institute for Solid State Physics and Optics, Wigner RCP, 1525 Budapest, Hungary; 3Pediatric Center, Semmelweis University, 1085 Budapest, Hungary; lengyel.anna1@med.semmelweis-univ.hu (A.L.); pinti.eva@med.semmelweis-univ.hu (É.P.); fekete.gyorgy@med.semmelweis-univ.hu (G.F.); 4Institute of Atomic Physics and Spectroscopy, University of Latvia, 1586 Riga, Latvia; ilze.lihacova@gmail.com (I.L.); aleksejs.lihacovs@gmail.com (A.L.); evplorina@gmail.com (E.V.P.)

**Keywords:** Fabry disease, angiokeratoma, dermoscopy, non-linear microscopy, multispectral imaging

## Abstract

Fabry disease (FD) is a multisystemic X-linked lysosomal storage disease that presents with angiokeratomas (AKs). Our objective was to investigate the clinical and morphologic features of AKs and to present two experimental techniques, multispectral imaging (MSI) and non-linear microscopy (NLM). A thorough dermatological examination was carried out in our 26 FD patients and dermoscopic images (*n* = 136) were evaluated for specific structures. MSI was used for the evaluation of AKs in seven patients. NLM was carried out to obtain histology samples of two AKs and two hemangiomas. Although AKs were the most common manifestation, the majority of patients presented an atypical distribution and appearance, which could cause a diagnostic challenge. Dermoscopy revealed lacunae (65%) and dotted vessels (56%) as the most common structures, with a whitish veil present in only 25%. Autofluorescence (405 nm) and diffuse reflectance (526 nm) images showed the underlying vasculature more prominently compared to dermoscopy. Using NLM, AKs and hemangiomas could be distinguished based on morphologic features. The clinical heterogeneity of FD can result in a diagnostic delay. Although AKs are often the first sign of FD, their presentation is diverse. A thorough dermatological examination and the evaluation of other cutaneous signs are essential for the early diagnosis of FD.

## 1. Introduction

Fabry disease (OMIM301500) is a pan-ethnic, X-linked inherited disorder of glycosphingolipid metabolism, affecting both hemizygous males and to a certain degree, heterozygous females. Prevalence of Fabry disease was originally estimated to be 1 in 40,000 males [1], but newborn screening studies indicate a higher reported incidence between 1:3100–1:1250, with it being the most frequent lysosomal storage disease [2,3]. Newborn screening in Hungary indicated an incidence of 1:13,341 [4]. The reason for this discrepancy in prevalence and incidence is suggested to be due to the clinical heterogeneity of Fabry disease [2]. Fabry disease is caused by mutations in the *alpha-galactosidase A* gene (*GLA*). The consequential impairment of alpha-galactosidase A activity leads to the lysosomal accumulation of globotriaosylceramid (Gb3) in different tissues and as a consequence, due to the progressive damage of the affected organs, diverse clinical symptoms develop. The clinical picture of Fabry disease is characterized by the progressive development of both cutaneous and extracutaneous signs and symptoms. Patients with minimal or no alpha-galactosidase A activity show the “classic” phenotype, with the early onset of characteristic symptoms including angiokeratoma corporis diffusum universale (ACDU), acroparesthesia, and cornea verticillata. From the third decade of life onward, renal, cardiovascular, and cerebrovascular disorders develop, some of which can have a considerable impact on quality of life [5,6]. Eventually, if left untreated, Gb3 accumulation in various cells leads to irreversible end-organ damage and is associated with reduced life expectancy [7]. In the case of atypical, late-onset variants, patients develop symptoms between the ages of 30 and 70 and usually present a milder phenotype. Late-onset variants often only affect one or a few organ systems, e.g., the cardiac variant and the renal variant [2]. The clinical presentation of Fabry disease in heterozygous females can range from asymptomatic to severe disease. However, females have a lower likelihood of developing the severe “classic” phenotype due to residual enzyme activity and the X-chromosome inactivation pattern [8]. The presence of a family history and/or the typical manifestations may lead to a suspicion of Fabry disease. Diagnosis in males is based on reduced alpha-galactosidase A activity in plasma, leukocytes, or other cell types. Although genetic analysis in males is not required for diagnosis, it is important for the selection of disease-specific therapies. Alpha-galactosidase A enzyme activity in females may be normal; therefore, a genetic analysis is necessary to confirm the diagnosis [9]. 

The cutaneous hallmark of Fabry disease is ACDU, which is the most frequent clinical manifestation of the disease that occurs in 66% of male and 36% of female patients [10,11]. ACDU is reported to develop in childhood in males, thus it is important for early diagnosis [11]. Angiokeratomas are acquired benign vascular malformations that are histologically characterized by dilated subepidermal vessels with overlaying epidermal acanthosis and hyperkeratosis. The angiokeratomas in Fabry disease develop due to the weakening of the vessel wall as a result of the endothelial accumulation of Gb3 in the dermis [12]. The lesions initially present as telangiectatic vessels, then develop into subepidermal vascular cavernae with occasionally warty hyperkeratosis presenting as solitary or clustered red-blue to black papules or plaques [13]. ACDU is characterized by multiple angiokeratomas, typically in a “bathing suit distribution” localized between the navel and the upper thighs [14]. Papules gradually appear over time and are typically more severe in the affected men [11]. Dermoscopy shows well-demarcated dark red to blueish black lacunae which correspond to dilated vascular spaces that can be thrombosed. Hyperkeratosis and acanthosis result in an opalescent whitish veil [15,16]. Other vascular lesions such as hemangiomas need to be considered in differential diagnosis [16]. The dermoscopic description of hemangiomas in the literature is almost identical to angiokeratomas, not emphasizing the minor differences in structure which render dermoscopic differentiation challenging. Additionally, while angiokeratomas are the most prominent signs of Fabry disease, it is noteworthy to mention that there are subtypes that are not associated with Fabry disease. Angiokeratomas can be classified into five distinct clinical types based on their location, morphology, and etiology, although they share similar histological features: (1) ACDU, as described earlier, presents in the bathing suit distribution in patients with Fabry disease, (2) angiokeratoma of Mibelli appears on the dorsal surfaces of the hands and feet and is often linked to chilblains, (3) angiokeratoma of Fordyce can be observed on the scrotum and is associated with increased venous pressure (4) angiokeratoma circumscriptum naeviforme manifests unilaterally on a lower extremity, and (5) solitary and multiple angiokeratomas, which can appear anywhere on the body but are most commonly found on the lower extremities [13]. It is noteworthy to mention that, while diffuse angiokeratomas are a hallmark feature of Fabry disease, they can occur in other lysosomal storage diseases including fucosidosis, sialidosis, GM1 gangliosidosis, galactosialidosis, beta-mannosidosis, Schindler disease type II, and aspartylglucosaminuria [5,11]. Apart from ACDU, other dermatological signs of Fabry disease include telangiectasias, abnormalities of sweating, lymphedema, and typical pseudoacromegaloid dysmorphia of the face or ‘facies Fabry’ [17]. 

Currently, there are two forms of enzyme replacement therapies available, agalsidase-alfa and agalsidase-beta, both administered as a biweekly intravenous infusion. A newer therapeutic approach is chaperone therapy with migalastat, which improves protein folding and the stability of the mutated endogenous alpha-galactosidase A protein but can only be used in specific missense mutations. Future therapies include next-generation enzyme replacement therapy, substrate reduction therapies, and gene therapy [7]. Initiation of disease-specific treatment is essential for delaying or preventing the progression of organ damage and improving life expectancy [9]. Even though enzyme replacement therapy has been shown to decrease Gb3 accumulation in the skin, based on clinical evidence, enzyme replacement therapy alone does not considerably improve skin status [17,18]. Therefore, for the optimal management of angiokeratomas in Fabry disease, apart from disease-specific treatment, destructive local therapies with vascular lasers should be implemented. The high variability of clinical manifestations could lead to a diagnostic delay which consequently impedes the initiation of disease-specific treatment [19]. Characteristic dermatological manifestations are frequent and could aid in establishing the diagnosis, thus the role of the dermatologist in the diagnosis of Fabry disease is essential.

The objective of this study was to investigate the clinical and morphologic features of ACDU in a group of Hungarian Fabry patients examined at Semmelweis University, Department of Dermatology, Venereology, and Dermato-oncology. In addition to clinical and dermoscopic examination, we present two experimental devices, multispectral imaging and ex vivo non-linear microscopy, for the examination of angiokeratomas in Fabry disease. Multispectral imaging is a promising in vivo imaging technique that uses different wavelengths to capture images, providing a map of skin chromophores including melanin and hemoglobin [20,21]. Non-linear microscopy is a novel label-free imaging method that can provide high-resolution imaging of the skin [22]. Keratins, elastin, and melanin can be visualized by two-photon absorption fluorescence (TPF), while second-harmonic generation (SHG) can be used for the imaging of collagen [23].

## 2. Materials and Methods

### 2.1. Patients

Here, we report on 26 patients from 16 families that were referred to our hospital and diagnosed with Fabry disease. The diagnosis of Fabry disease was based on clinical findings, decreased alpha-galactosidase A activity, and sequencing of genomic DNA. A thorough dermatological examination was carried out including dermoscopic assessment and photographic documentation. In the case of two patients, one male and one female, the diagnosis of angiokeratoma was also confirmed by histopathology. Clinical data concerning the extracutaneous manifestations of Fabry disease were acquired retrospectively. Informed consent was obtained from all patients included in the study.

### 2.2. Dermoscopy

Dermoscopic (Heine Delta 20, Heine Optotechnik GmbH, Herrsching, Germany) images of randomly selected target lesions were evaluated for the presence of specific features such as red, blue, or black lacunae and the presence of a whitish veil. The structure of vasculature was also evaluated. In this work, we used descriptive as well as frequently used metaphoric dermoscopic terms [3,16].

### 2.3. Multispectral Imaging

A handheld prototype has been used as described earlier [24,25], developed by the University of Latvia in collaboration with Riga Technical University (Riga, Latvia). A set of autofluorescence images under continuous 405 nm LED excitation was recorded. In addition, diffuse reflectance images were acquired under 526 nm, 663 nm, and 964 nm illumination. Four battery-powered violet and green LEDs were placed within a cylindrical light-shielding wall that also ensured a fixed distance (60 mm) between the camera and the evenly illuminated skin. A long pass filter (>515 nm) was placed in front of a color CMOS 5-megapixel IDS camera (MT9P006STC, IDS uEye UI3581LE-C-HQ, Obersulm, Germany) to prevent detection of 405 nm LED emission. Here, we assessed the angiokeratomas of 7 patients in our patient cohort.

### 2.4. Non-Linear Optical Microscopy

Deparaffinized and unstained sections from the angiokeratoma biopsies were prepared for non-linear microscopic investigations. Non-linear microscopic imaging was performed using a ~20 MHz repetition rate, a sub-ps Ti:Sapphire laser (FemtoRose 300 TUN LC, R&D Ultrafast Lasers Ltd., Budapest, Hungary), and an Axio Examiner LSM 7 MP laser-scanning 2P microscope with a 20×, 1.0 NA water immersion objective (W-Plan-APOCHROMAT, Carl Zeiss Microscopy GmbH, Jena, Germany). The central wavelength of the pump laser was set to 800 nm, with a bandwidth of <2 nm. A 405/20 nm bandpass filter was used to collect the SHG signal (collagen) and a 525/50 nm (green) bandpass filter was collecting the TPF signal before the NDD detectors. The green channel was chosen based on preliminary measurements on blood. Two-channel, 16-bit images were captured from individual imaging areas of 420 × 420 μm^2^ to map the entire slide. The pixel dwell time was set to 12 μs. The acquired TPF and SHG images were merged and assembled into two-channel images with ImageJ v1.46 software (NIH, Bethesda, MD, USA). Conventional hematoxylin eosin staining was carried out from different sections of the same lesion as a control.

## 3. Results

### 3.1. Diagnosis and Treatment

Our patient group consisted of 12 males and 14 females who range in age from 5 to 70 years, with a mean age of 38 ± 20.8 years. The age at diagnosis of Fabry disease ranged between 4 and 60 years with a mean age of 32.8 ± 19.4 years (Table 1). Overall, 16 patients were diagnosed as a result of an affected family member. In the other part of our patient cohort, characteristic organ manifestations led to the clinical suspicion of Fabry disease, most commonly ACDU, in four patients. Disease-specific treatment was introduced in the majority of patients shortly after the diagnosis was confirmed (Table 1). Patients who at this point do not receive the specific therapy present no early signs of organ involvement, for which they are screened regularly.

### 3.2. Extracutaneous Manifestations

Cornea verticillata of the eye was the most common extracutaneous manifestation concerning 18 patients, each followed by cardiac involvement and acroparesthaesia. Eleven patients presented renal involvement that varied from mild proteinuria to organ failure necessitating dialysis. Pulmonological manifestations were rare and mostly subclinical, resulting in restrictive or mixed restrictive-obstructive ventilatory impairment (Table 2).

### 3.3. Cutaneous Findings

We found angiokeratomas in the majority of our patient group, including 13 female and 11 male patients (Figure 1). One young male and one young female patient had no angiokeratomas upon thorough clinical examination. The characteristic ACDU in the bathing suit distribution was present in eight males and two females, with angiokeratomas localized between the navel and the knees. However, in severe cases, ACDU also progressed towards the upper extremities. In our patient group, a wide spectrum of the characteristic ACDU phenotypes could be observed. Typically, male patients had extensive, hyperkeratotic, warty angiokeratomas (Figure 2A–C), while other patients presented with dark red to purple maculae corresponding to vessel ectasiae without overlying hyperkeratosis (Figure 2D–G,I). Here, we used the term incipient or macular angiokeratomas for the latter phenotype. It is worth mentioning that while warty angiokeratomas are easy to recognize, incipient lesions in some cases could be easily missed without dermoscopic examination. Two female patients presented with ACDU with warty angiokeratomas in the bathing suit distribution (Figure 2H). In contrast, the majority of female patients had fewer and less conspicuous angiokeratomas (Figure 2I,J).

Overall, the most frequent localization of angiokeratomas in our patient cohort was on the chest, followed by the abdomen and umbilical region (Figure 3A). Skin sites associated with the bathing suit distribution, including the umbilical region, gluteal region, thighs, and genitalia were more commonly affected in males, whereas the chest, upper extremities, and upper back were more frequently involved in female patients. We observed angiokeratomas in acral localizations including the palmoplantar region, ears, and lips (Figure 2F,G,I). Mucosal involvement was rare. We found no angiokeratomas on the back of the hands in our patient group. Here, we distinguished between the typical and atypical distribution of angiokeratomas, with typical referring to the bathing suit distribution. In our patient group, males were most likely to present with angiokeratomas in the typical bathing suit distribution, while more females presented with angiokeratomas on atypical skin sites such as the upper trunk and arms (Figure 3B). While angiokeratomas appeared in groups in the typical bathing suit distribution, in the case of the atypical distribution, solitary lesions were characteristic. In the case of the atypical distribution, the majority of patients, three males and eight females, had only a few numbers (*n* < 5) of angiokeratomas. Three females in the atypical distribution group presented with more angiokeratomas, including two middle-aged women who had several solitary papular lesions on the trunk and upper extremities and one young female patient who presented diffuse incipient angiokeratomas only on the upper arms. This shows that only ten of our patients showed the “classic” ACDU phenotype, other cases were atypical regarding distribution and clinical appearance and were not clinically conspicuous.

Other cutaneous manifestations included anhidrosis or hypohidrosis in 11 patients, additionally, seven patients presented with hyperhidrosis (Figure 1). Dry skin was also common in association with reduced sweating. Other vascular lesions were also found in the majority of our patients, including telangiectasias and cherry angiomas. Almost half of the patients in our group showed pseudoacromegaloid features to some extent. Interestingly, varicose veins were common in our patient population and could also be observed on young patients (Figure 4A,B). Lymphoedema was also seen in five adult patients (Figure 4C). Synophris was also observed in four young patients, both males and females. It is important to note that patients with only a few angiokeratomas or without angiokeratomas all showed other specific cutaneous signs that could aid in the establishment of the diagnosis.

### 3.4. Dermoscopic Features

Upon dermoscopic examination of the angiokeratomas, we observed the vascular structure, the colors, and the presence of the whitish veil (Figure 5). We examined 135 dermoscopic images of angiokeratomas. Here, we classified vascular structures as glomerular, lacunar, dotted, and linear-irregular. Glomerular vessels were present in 30% of the cases, almost two-thirds of the cases presented with lacunae, and more than half of the cases presented with dotted vessels. The coincidence of lacunae and dotted vessels was quite common, 44% of the examined cases showed both features. A whitish veil corresponding to the overlaying hyperkeratosis was present in only 25% of the examined cases. Structureless milky-red areas were present in 26% of the cases, where neither vascular structures nor the hyperkeratotic whitish veil could be observed. Two-thirds of the lesions were a dark purple or red color and in 18% of cases, the vasculature appeared bright red, while in 16% of the cases, both colors were present in a lesion. Upon dermoscopic examination, palmoplantar and intraumbilical lesions showed a monomorphic appearance across patients with dotted vessels and small lacunae (Figure 5G–I). Due to the monomorphic appearance of these sites, these images were excluded from the detailed evaluation above of the presence of dermoscopic structures. In addition to angiokeratomas, other vascular malformations were common including hemangiomas and telangiectasias (Figure 5J–L). Hemangiomas could be distinguished from angiokeratomas by the proliferative capillaries and white septae.

### 3.5. Non-Linear Microscopic Imaging

Skin biopsies of angiokeratomas in Fabry patients were used for non-linear microscopic evaluation, while a hemangioma biopsy sample was used for differential diagnosis purposes. Formalin-fixed, previously paraffin-embedded, deparaffinized sections were assessed with TPF and SHG as well, where 2D vertical cross-section mosaic images were acquired to mimic conventional histology. The SHG signal visualized in a magenta color showed collagen fibers in the dermis. The TPF channel in green showed the keratinocytes of the epidermis, while the lamellar structures in green represent the stratum corneum in both samples. Morphologic differences could be observed, including hyperkeratosis and dilated vessels in the epidermis in the case of the angiokeratoma. In the case of the hemangioma, hyperkeratosis was absent and the vasculature was not limited to the epidermis and papillary dermis. The vessel morphology was also different with smaller, proliferative capillaries (Figure 6A–D). Figure 6E,F shows the control conventional histopathologic images of the same lesions.

### 3.6. Multispectral LED Imaging

Using multispectral diffuse reflectance and autofluorescence imaging, angiokeratomas appear as low-intensity areas compared to the surrounding healthy skin which is most noticeable in the case of the green autofluorescence channel. It is noteworthy that in the case of the autofluorescence channel especially, but also on the green channel, the subepidermal vasculature in the background can be more easily distinguished compared to dermoscopic images. In the context of papular angiokeratomas, using infrared illumination, the areas of the lesions exhibit reduced diffuse reflectance signals. In comparison, macular angiokeratomas do not show low diffuse reflectance under the same conditions (Figure 7). 

## 4. Discussion

Fabry disease is the most common lysosomal storage disease that affects multiple organ systems. Without treatment, it can lead to end-stage organ damage and premature death [2,5]. The clinical heterogeneity of the disease can lead to diagnostic delays that set back disease-specific treatment.

Cutaneous involvement is common and usually represents an early manifestation, thus the correct evaluation of cutaneous signs is essential for the early suspicion of Fabry disease. For this reason, the dermatologist’s role is essential in the multidisciplinary Fabry team. In our patient group, angiokeratomas were more frequent, presenting in 92% of our patients compared to the literature where 69% of patients presented with angiokeratomas [8]. Here, we found that angiokeratomas are more likely present in the bathing suit distribution in males, while the most frequent localizations for angiokeratomas in females are on the trunk and extremities, which coincide with the literature [5]. However, based on our findings, only 38% of our patients, mainly males, showed characteristic ACDU as clustered angiokeratomas in the bathing suit distribution. The other almost two-thirds of the patients either had no or very few angiokeratomas or presented with atypical angiokeratomas in distribution and appearance. This highlights that angiokeratomas in Fabry disease show a very diverse presentation beyond the classic presentation of ACDU, meaning that only a little more than one-third of our patient group presented the clinically unambiguous phenotype of ACDU while the majority of the cases were not conspicuous and could pose a diagnostic challenge. Additionally, regardless of distribution, incipient angiokeratomas are less visible and could potentially be overlooked without thorough dermatological examination or by less experienced examiners, thus causing a diagnostic delay. We would like to highlight that angiokeratomas in Fabry disease show a very diverse presentation beyond the typical presentation of ACDU. On the other hand, while the majority of our patient group presented as atypical ACDU, it is important to note that this group presented with other cutaneous signs of Fabry disease, e.g., hypohidrosis. Therefore, the constellation of these less specific cutaneous signs among the extracutaneous manifestations could aid in establishing the diagnosis even in the absence of the typical ACDU. Here, we observed synophris in four cases and would like to present it as a new sign of Fabry disease. While synophris can appear as a normal variation in different ethnicities [26], it can also be a sign of other lysosomal diseases [27]. Additionally, varicose veins could be observed in almost half of our patient group, affecting younger patients as well. To the best of our knowledge, only two case studies have mentioned varicose veins in Fabry disease. One of the cases presented bilateral leg ulcers at an early age where venography revealed venous reflux and varicose veins, and further exploration of the family history led to the diagnosis of Fabry disease [28]. The other case report merely noted the presence of lower extremity varicose veins [29]. While the effects of Fabry disease on venous tissue are not well studied, we hypothesize that venous insufficiency in the lower extremities might be a common but overlooked manifestation of Fabry disease. 

To date, more than 1000 mutations of the *GLA* gene have been identified [30]. There are initiatives attempting to establish genotype-phenotype correlations, e.g., the classic phenotype and late-onset phenotypes including cardiac and renal variants. However, there are still no phenotypic classifications for many GLA variants [31]. Although the genotype-phenotype correlation was out of the scope of this article, morphologic similarities could be observed across family members in our patient group. For example, a mother and her son and a family of middle-aged brothers showed only a small number of angiokeratomas (<5) in an atypical distribution. In another family of two middle-aged brothers, we only observed incipient angiokeratomas in the bathing suit distribution where despite their age, the angiokeratomas were discrete. Overall, considering the clinical heterogeneity in our patient group, intrafamilial similarities could be observed.

With novel therapies, a need for monitoring efficacy arose and skin manifestations, due to easy accessibility, could be a potential target for evaluation of treatment. Although after five months of enzyme replacement therapy, the superficial capillary endothelium in the skin showed complete clearance of Gb3 via electron microscopy [32], complete resolution of angiokeratomas should not be expected. However, there have been examples where angiokeratomas showed remission after long-term therapy [13,33].

Here, we examined 135 dermoscopic images of angiokeratomas among the 24 patients in our cohort who presented with angiokeratomas, which to our knowledge is the most extensive in the literature. Angiokeratomas presented most commonly as dark red lacunae and the coincidence of dotted vessels was also common. Interestingly, here we found that the whitish weil was only present in one-quarter of the cases, whereas the literature reported an incidence of 77% [34]. However, it is important to note that dermoscopy of these lesions showed high variability, especially in the case of atypical solitary angiokeratomas where glomerular and irregular vascular structures were more common. Dermoscopy proved to be a valuable tool for the examination of incipient macular angiokeratomas and the examination of the navel.

Multispectral imaging is an emerging modality in skin cancer diagnostics as well as for rare diseases [21,23,35]. Here, the utilized multispectral imaging device showed fewer details compared to dermoscopy, but the background vasculature could be more precisely distinguished, especially in the autofluorescence channel due to the absorption of hemoglobin. Under infrared illumination, in the case of papular angiokeratomas, the lesion areas show low diffuse reflectance signals that may correspond to the deeper extent of the lesions. Similar observations were made in the case of melanomas with different invasion depths [20]. Multispectral imaging enables imaging that is selective for certain molecules present in hemangiomas and angiokeratomas, such as hemoglobin and keratin. Therefore, compared to dermoscopy using visible white light, additional data on lesion depth and underlying vasculature can be collected. While dermoscopy and multispectral imaging provide a summary of different signals, non-linear microscopy is a label-free imaging method that provides optical sectioning and molecular selectivity with a tissue resolution similar to conventional histology. Ex vivo non-linear microscopy has been used successfully in the diagnosis of basal cell carcinoma [36] and other rare inherited skin diseases, for example, pseudoxanthoma elasticum [37] and Ehlers-Danlos syndrome [38]. Here, we could differentiate angiokeratomas from hemangiomas based on morphologic signs similar to conventional hematoxylin-eosin staining. In our case, vascular spaces appeared as black lumens due to the preparation procedure. The use of fresh frozen samples could be used to model a possible in vivo imaging method, where the tissue is not potentially modified from the paraffination and deparaffination procedure and the vascular spaces are visualized by hemoglobin on the green channel. On the other hand, both techniques require special imaging devices that are not available in standard clinical practice at the moment and are mostly prototypes. The cost of multispectral imaging is similar to that of dermoscopy, while non-linear microscopy is an expensive technique, even exceeding the costs of reflectance confocal microscopy. Therefore, non-linear microscopy is mainly used for research purposes. It is noteworthy to mention that a limitation of non-linear microscopy is that, during non-linear microscopy imaging, the TPF signal of certain molecules can overlap. In addition, during multispectral imaging, autofluorescence signals and diffuse reflectance signals generated by LED excitation originate from different tissue layers, potentially causing interference.

Angiokeratomas are often the first sign of Fabry disease and are significant features in the establishment of the diagnosis. Diffuse angiokeratomas, especially in the typical bathing suit distribution, should alert the physician to the necessity of further investigation into the possibility of Fabry disease or other storage diseases. However, as seen in our patient group, especially in females, angiokeratomas can possess atypical distributions affecting other skin sites and present as solitary lesions rather than in groups, thus making diagnosis challenging. We would like to emphasize the importance of dermoscopy, as incipient angiokeratomas or intra-umbilical angiokeratomas could be easily missed without a thorough dermoscopic examination. Novel optical modalities such as multispectral imaging and non-linear microscopy could possibly aid in the diagnosis of atypical cases. 

## Figures and Tables

**Figure 1 diagnostics-13-02368-f001:**
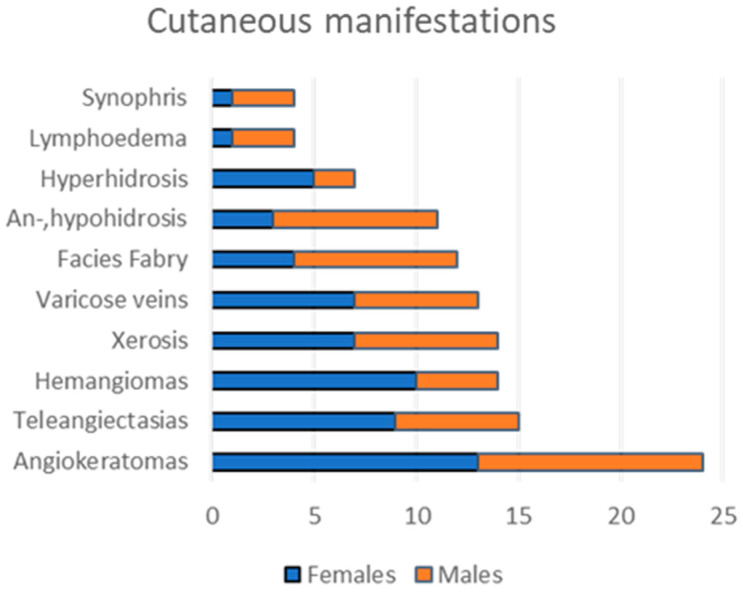
Cutaneous manifestations in our patient group (*n* = 26). Angiokeratomas were the most frequent cutaneous findings, followed by other vascular lesions including telangiectasias and hemangiomas. Note that here we observed varicose veins in half of our patient group, and we also found synophris in 4 patients.

**Figure 2 diagnostics-13-02368-f002:**
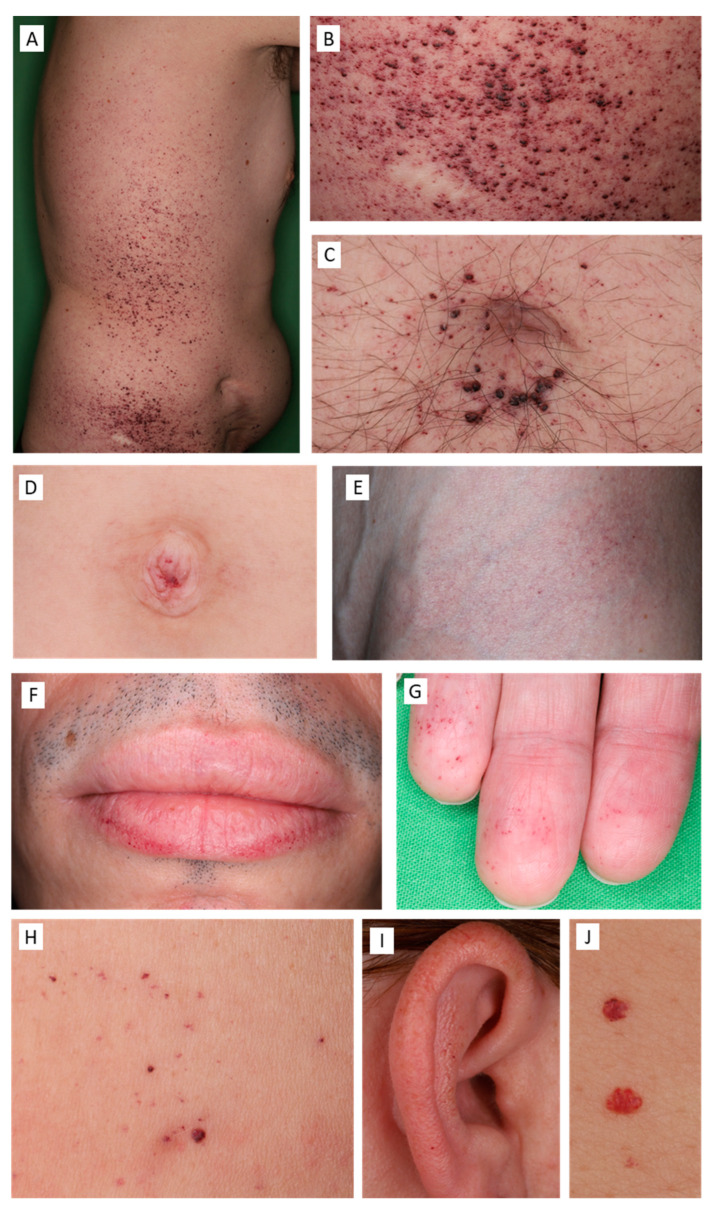
Angiokeratomas in Fabry disease in male patients (**A**–**G**) and female patients (**H**–**J**). (**A**–**G**): Angiokeratoma corporis diffusum universale. (**A**–**C**): warty, papular angiokeratomas on the trunk (**A**), inguinal region (**B**), and umbilical region (**C**). (**D**–**G**): incipient, macular angiokeratomas in the umbilical region of a young male patient (**D**), inguinal region (**E**), on the lips (**F**), and on the fingertips (**G**). Angiokeratoma corporis diffusum universale with warty angiokeratomas on the lower abdomen of a female patient (**H**), macular angiokeratomas on the helix and anthelix of a female patient (**I**), solitary angiokeratomas on the chest of a female patient (**J**).

**Figure 3 diagnostics-13-02368-f003:**
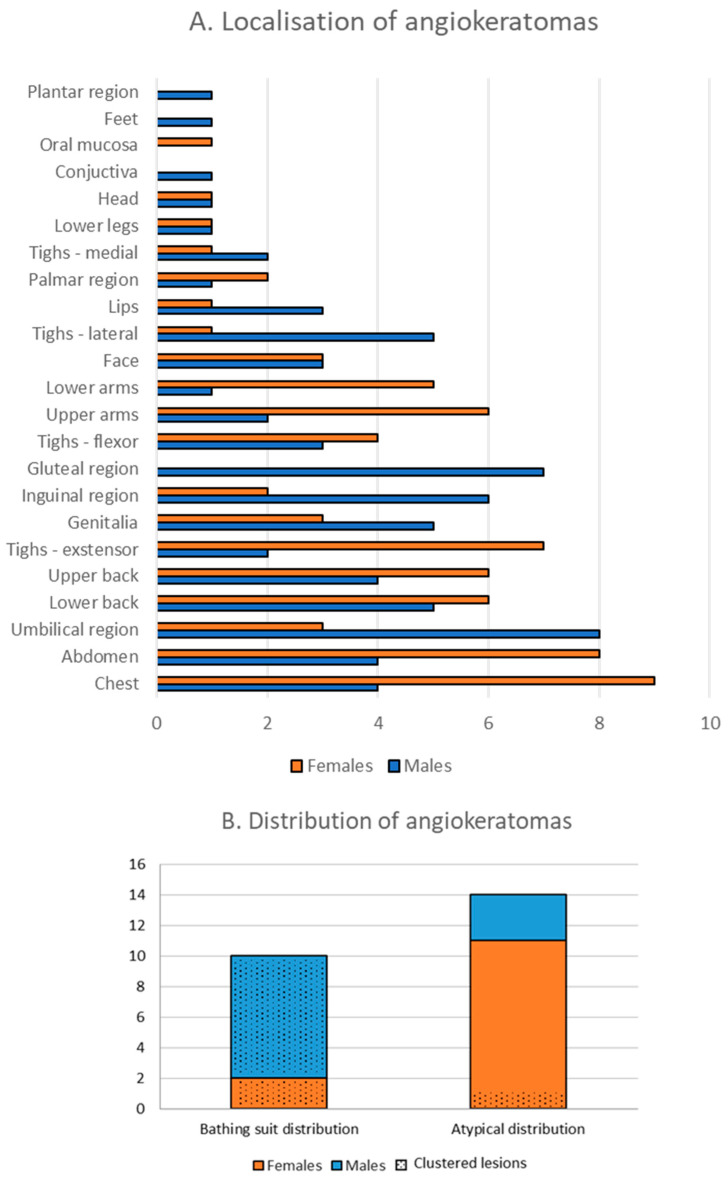
(**A**): Frequency of angiokeratomas on different skin sites in males and females in our patient cohort. Note that the chest is the most common localization of angiokeratomas with a female dominance. The umbilical, gluteal, and inguinal regions are most common among males, respectively, corresponding to the bathing suit distribution. (**B**): Distribution of angiokeratomas in males and females. Dotted columns indicate patients presenting with grouped angiokeratomas, while solid colors represent patients that presented with solitary lesions.

**Figure 4 diagnostics-13-02368-f004:**
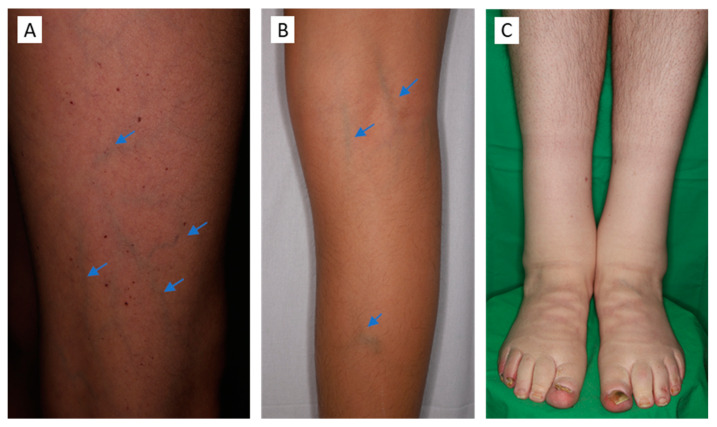
(**A**,**B**): Varicose veins (blue arrows) on two young male patients. The young male patient in panel (**A**) also presented with several angiokeratomas. (**C**): Lymphoedema in a middle-aged male patient.

**Figure 5 diagnostics-13-02368-f005:**
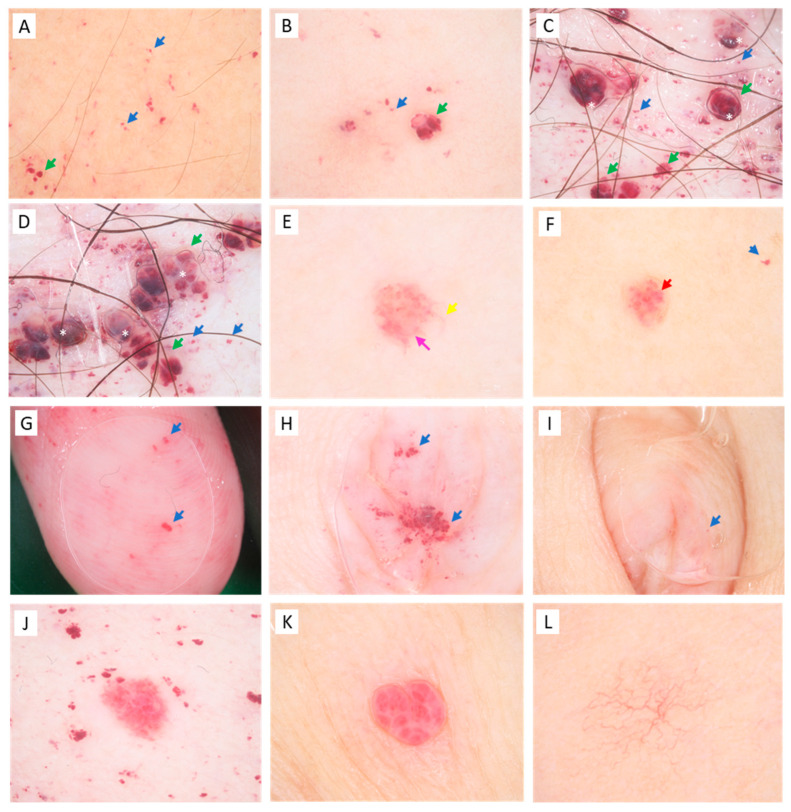
Dermoscopic images of vascular structures. (**A**): group of macular angiokeratomas on a young male patient, (**B**–**D**): groups of macular and papular angiokeratomas on a female (**B**) and male patient (**C**), (**E**,**F**): solitary angiokeratomas on female patients, (**G**): macular angiokeratomas on the fingertips of a male patient. Blue arrows represent dot-like vessels, green arrows show lacunar structures, white asteriks indicate the whitish veil, red arrows indicate glomerular vessels, the pink arrow shows a structureless milky-red area, and the yellow arrow shows a linear-irregular vessel. (**H**,**I**): macular angiokeratomas in the umbilicus of two young male patients. (**I**): Note the discrete vascular ectasiae that were barely visible without dermoscopic imaging. (**J**): hemangioma among angiokeratomas on a male patient (**K**): hemangioma on a female patient. Note the fibrous septae and bright red lacunae in the case of the hemangiomas (**J**,**K**). (**L**): Spider nevus on a young male patient.

**Figure 6 diagnostics-13-02368-f006:**
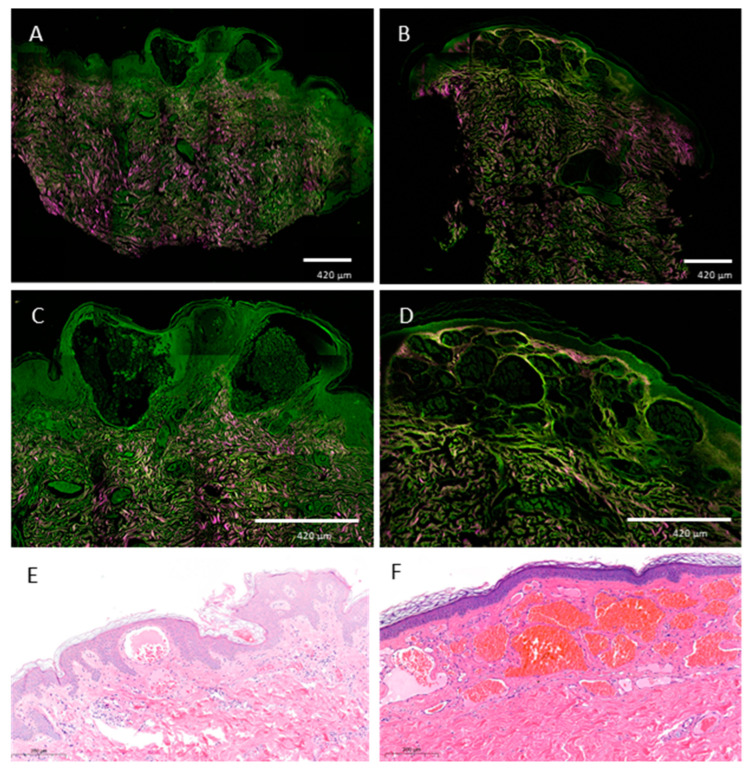
Morphological features of an angiokeratoma and a hemangioma visualized by ex vivo non-linear microscopy. Non-linear microscopic images of an angiokeratoma (**A**,**C**) from our patient group and a hemangioma as a differential diagnosis (**B**,**D**). Note the hyperkeratotic surface and the dilated capillaries that are limited to the papillary dermis in the angiokeratoma sample (**A**,**C**), while in the hemangioma, capillaries extend further towards the subcutaneous tissues. Concerning the vessel morphology, angiokeratomas present with ectatic, dilated vessels (**C**). By moving downward, normal vessel morphology of the dermis can be seen. In the case of the hemangioma sample, the morphology suggests a proliferative process with vessels smaller in diameter divided by thin fibrous septae (collagen fibers in pink) that extend beyond the papillary dermis (**D**). Hematoxylin-eosin staining of an angiokeratoma (**E**) and hemangioma (**F**). The angiokeratoma features dilated vessels with acanthosis (**E**). The hemangioma presents with multiple vascular spaces and normal epithelium (**F**). Scale bar: 420 μm (**A**–**D**) 200 μm (**E**,**F**).

**Figure 7 diagnostics-13-02368-f007:**
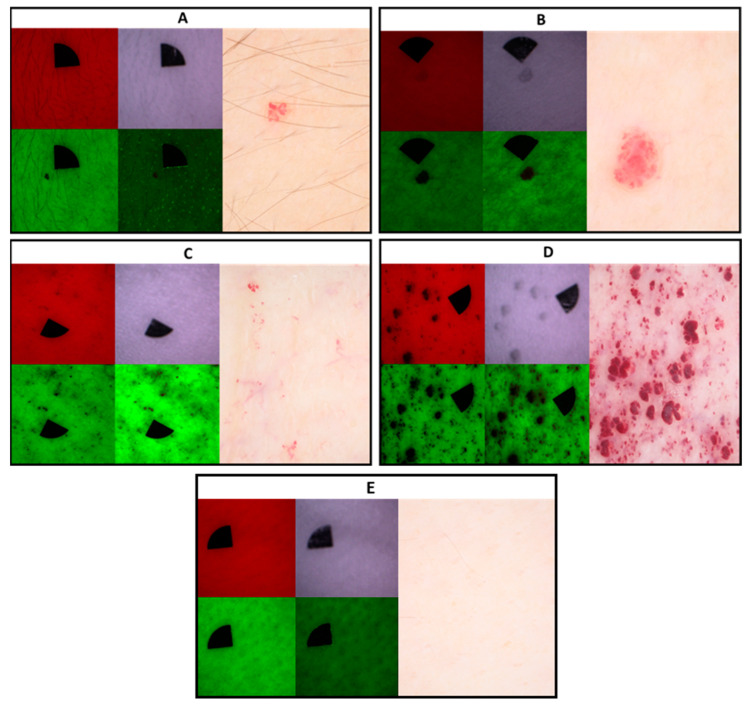
Representative images of angiokeratomas with different morphologies using multispectral imaging in comparison to dermoscopic images of the selected lesions. Panel (**E**) represents normal skin. (**A**): solitary macular angiokeratoma in a young male patient, (**B**): solitary papular angiokeratoma of a female patient, (**C**): clustered macular angiokeratomas in a male patient and (**D**): clustered papular and macular angiokeratomas in a male patient. (**E**): normal skin without angiokeratomas on a healthy individual. Each panel contains diffuse reflectance images at 663 nm (red light, upper left image), 526 nm (green light, lower left image), 964 nm (infrared light, upper right image), and autofluorescence images at 405 nm (lower left pictures) LED excitation. Dermoscopic images are on the far-right side of each panel. Note the prominent underlying vasculature in the case of the clustered lesions under 526 nm and 405 nm illumination (panel (**C**,**D**) bottom row). Papular lesions appear as low-intensity areas under infrared illumination (panel (**B**,**D**) upper right images).

**Table 1 diagnostics-13-02368-t001:** Clinical details of Fabry patients in our cohort.

	Total (*n* = 26)	Females (*n* = 14)	Males (*n* = 12)
Mean age (years)	37.0 ± 20.9	42.1 ± 19.9	31.1 ± 21.7
Mean age at diagnosis (years)	32.77 ± 19.4	37.07 ± 17.4	27.75 ± 21.1
Mean age at the initiation of therapy (years)	36.1 ± 18.3	43 ± 15.1	29.70 ± 19.4
Therapy			
Agalsidase-α	10 (38%)	5 (19%)	5 (19%)
Agalsidase-β	8 (31%)	4 (15.5%)	4 (15.5%)
Migalastat	1 (4%)	0 (0%)	1 (4%)
None	7 (27%)	5 (19%)	2 (8%)

**Table 2 diagnostics-13-02368-t002:** Extracutaneous manifestations in our patient group.

	Total (*n* = 26)	Females (*n* = 14)	Males (*n* = 12)
**Cardiovascular involvement**			
Cardiac involvement	17	9	8
Hypertension	10	6	4
**Cerebrovascular involvement**			
Headaches	4	1	3
Tinnitus	1	0	1
Vertigo	1	1	0
Stroke	4	3	1
Hearing impariment	5	2	3
**Renal involvement**	11	6	5
**Acroparesthaesia**	16	6	10
**Opthalmologic signs**			
Cornea verticillata	18	9	9
Other nonspecific signs	14	8	6
Gastrointestinal involvement	11	4	7
Pulmonary involvement	4	3	1
Fatigue	13	7	6

## Data Availability

Not applicable.

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
