# Peer review of "A Cross-Sectional Study of the Dermatological Manifestations of Patients with Fabry Disease and the Assessment of Angiokeratomas with Multimodal Imaging"

_diagnostics, 2023, doi:10.3390/diagnostics13142368_

Round 1
Reviewer 1 Report
Dear authors,
thank you for your informative and well-written article. I would recommend to improve only a few points:
1) could you highlight why you did additionally MSI and NLM and what is the benefit compared to dermoscopic imaging
2) what are the limitations of MSI and NLM (application in daily practice, costs?)
3) do you have correlation between NLM and goldstandard histology? Maybe you can add the corresponding H&E histologic images to Figure 6 which makes it more clear for colleagues who are not that familiar with the NLM imaging procedure.
Thank you and kind regards
Author Response
1.) We thank the Reviewer for this remark. Accordingly, we have included the following text in the revised manuscript:
“Multispectral imaging enables imaging that is selective for certain molecules present in hemangiomas and angiokeratomas, such as haemoglobin and keratin. Therefore, compared to dermoscopy using visible white light, additional data on lesion depth and underlying vasculature can be collected.”
“Non-linear microscopy is a label-free imaging method that provides optical sectioning and molecular selectivity with a tissue resolution similar to conventional histology. ”
2.) The authors thank the Reviewer for this important comment. In the revised manuscript, the following limitations have been included:
”On the other hand, both techniques require special imaging devices that are not available in the standard clinical practice at the moment and are mostly prototypes. The cost of multispectral imaging is similar to that of dermoscopy, while non-linear microscopy is an expensive technique, even exceeding the costs of reflectance confocal microscopy, thus non-linear microscopy is mainly used for research purposes. It is noteworthy to mention among limitations, that during non-linear microscopy imaging, TPF signal of certain molecules can overlap. Also, during multispectral imaging, autofluorescence signals and diffuse reflectance signals generated by LED excitation originate from different tissue layers, potentially causing interference.”
3.) The authors thank the reviewer for this valuable remark. We added the corresponding H&E stained images (Figure 7. E-F) from the same patients as the NLM evaluation to show correlation with histology.
Reviewer 2 Report
This is a potentially interesting paper studying on angiokeratomas inFabry disease. I was very interested in this paper initially thinking about using the AK as clinical endpoint but it is not clear that this was the goal of this paper. The paper can be improved in the following manner (not in a particular order):
1. The images and technique should be explained including what exactly are we seeing in each image. Also, normal control skin should be added next to each type of image so we can appreciate what the abnormality in the patients is. I don't understand what are lacunae, dots and whitish veil lesions.
2. Diminish the number of acronyms as much as possible.
3. The authors described AK in the majority of female patients. In my experience AK in women is rare and the abnormalities that are found are often confused with small angiomas. Therefore, the authors should dedicate a figure to AK in females, explaining what we see is, what the abnormality is and again with normal controls.
4. What are atypical AK in form and distribution? How can the majority of patients have atypical lesions?
5. The authors should add that AK are not specific for Fabry disease and can be seen in other lysosomal disease (e.g. mannosidosis, fucosidosis) and in non-lysosomal disorders.
6. AK are not seen in early childhood but only in the middle of the second decade of life.
7. What are AK capillary malformation? Clearly they develop over time, that is they are acquired and they can regress with appropriate therapy. They are acquired lesions, not malformation.
8. Please provide patients GLA mutations and enzyme activity in the males.
9. Figures, such as Fibure 2, please state if the sex of the patients.
10. Please state clearly what the advantages of these photography techniques are.
No comment
